# Intracellular and Tissue Levels of Vitamin B12 in Hepatocytes Are Modulated by CD320 Receptor and TCN2 Transporter

**DOI:** 10.3390/ijms22063089

**Published:** 2021-03-17

**Authors:** Joseph Boachie, Antonysunil Adaikalakoteswari, Ilona Goljan, Jinous Samavat, Felino R. Cagampang, Ponnusamy Saravanan

**Affiliations:** 1Division of Metabolic and Vascular Health, Warwick Medical School, University of Warwick, Coventry CV2 2DX, UK; J.Boachie@warwick.ac.uk (J.B.); J.Samavat@warwick.ac.uk (J.S.); 2Department of Biosciences, School of Science and Technology, Nottingham Trent University, Nottingham NG11 8NS, UK; 3Diabetes Centre, George Eliot Hospital NHS Trust College Street, Nuneaton CV10 7DJ, UK; ilga@novonordisk.com; 4Institute of Developmental Sciences, University of Southampton, Faculty of Medicine, Southampton General Hospital, Southampton SO16 6YD, UK; f.cagampang@soton.ac.uk; 5Division of Health Sciences, Warwick Medical School, University of Warwick, Coventry CV2 2DX, UK

**Keywords:** vitamin B12 (cobalamin), transcobalamin, hepatocytes, intrinsic factor, B12 receptor (CD320), B12 transporter (transcobalamin II, TCN2)

## Abstract

The liver mass constitutes hepatocytes expressing receptors for vitamin B12 (B12)-bound transporters in circulation. However, intrahepatic and circulating B12 interrelationship levels remain unclear. We assessed the intracellular B12 levels at various circulating B12 concentrations in human HepG2 cell-line and liver tissue levels of B12 in the C57BL/6 mouse model. In HepG2 cells treated with a range of B12 concentrations, the intracellular and circulatory B12 levels, transcript and protein levels of B12 receptor (CD320) and transporter (TCN2) were determined using immunoassays, qRT-PCR and Western blot, respectively. Similar assessments were done in plasma and liver tissue of C57BL/6 mice, previously fed a diet of either a high or low B12 (30.82 µg B12/kg and 7.49 µg B12/kg, respectively) for 8–10 weeks. The physiological B12 status (0.15–1 nM) resulted in increased levels of intracellular B12 in HepG2 cells compared to supraphysiological levels of B12 (>1 nM). Gene and protein expression of CD320 and TCN2 were also higher at physiological levels of B12. Progressively increasing extracellular B12 to supraphysiological levels led to relative decreased levels of intracellular B12, lower expression of gene and protein levels of CD320 and TCN2. Similar results were observed in liver tissue from mice fed on a low B12 diet verses high B12 diet. These findings suggest that unlike supraphysiological B12, physiological levels of B12 in the extracellular media or circulation accelerates active transport of B12, and expression of CD320 and TCN2, resulting in higher relative uptake of B12 in hepatocytes.

## 1. Introduction

Vitamin B12 (B12), also known as cobalamin (Cbl), is an essential water-soluble vitamin that is involved in cell metabolism. The main sources of B12 are animal products, supplements or fortified foods. B12 is difficult to produce chemically and is only naturally synthesized by fermentation with cobalamin-producing bacteria. Although some ruminating mammals absorb the B12 from the bacteria living in their gastrointestinal tract (GIT), humans cannot acquire the B12 from their enteric bacteria despite the large colonies, as they live primarily in a large intestine [1]. Intestinal uptake of B12 occurs in a small intestine, mainly aided by the binding of a gastric intrinsic factor (IF) with only 1% of free B12 known to be passively absorbed through the GIT [2]. Hence, humans rely mainly on the dietary sources of B12. However, the circulating levels of B12 are dependent on its receptors and transporters. Specific membrane receptors (CD320) and transport proteins (haptocorrin (HC) and transcobalamin II (TCN2)) facilitate the uptake and delivery of B12 to the liver (B12-HC) and other tissues (B12-TCN2) in humans [3]. Dietary uptake of B12 occurs by the sequential binding to HC and IF, followed by the uptake of B12-IF via specific ileal receptors (Cubilin) [4], then B12 is released into the bloodstream to form a B12-TCN2 complex, which becomes actively available for tissue uptake via the transcobalamin receptor CD320. Consequently, reduced intake or intestinal absorption of B12 can result in deficit levels of B12 tissue stores. Several studies show that B12 deficiency is associated with various metabolic and neurodevelopment diseases and is treated with oral supplementation.

TCN2 is the most essential protein in the delivery of B12 from the blood to various tissues and organs. The potential role of TCN2 in the transport of B12 from the mother to the foetus was reported in pregnant adolescents, where higher placental TCN2 expression was associated with higher cord blood B12 concentrations [5]. The transcobalamin receptor (CD320) is known to be expressed in actively dividing cells [6] and required for the uptake of TCN2-bound B12 [7]. Studies have also shown that CD320 knockout mice have B12 deficiency and have behavioural abnormalities, macrocytic anaemia, infertility and hypomethylation of DNA in the brain [8]. However, current knowledge on the absorption and tissue distribution of B12 has only been reported in few animal studies [9,10,11,12,13,14]. A study in Wistar rats showed that kidneys contained more B12 compared to other tissues such as lungs, spleen, brain, heart and liver [15]. B12 depleted rats have shown that the uptake of hydroxy-Cbl in the liver was more than that for cyano-Cbl, contrary to other tissues such as the kidney, brain and spleen [10]. Similarly, animals fed with imbalanced nutrients (low B12, high folate and high methionine) during pregnancy showed a redistribution of B12 in the maternal tissue from the kidneys to the liver and in the foetal compartment (uterus, placenta and foetuses) [16]. These studies evidenced that the receptors/transporters correlated to B12 levels and the differential tissue distribution depends on the B12 status. As TCN2/CD320 are the key transporter/receptor for B12 in many tissues, studying the levels of these two may likely provide an understanding of the relative cellular uptake of B12. Moreover, the role of receptor/transporter in modulating the intracellular levels of B12 at various levels of circulating B12 levels has not been explored. Liver is the main storage site and metabolic factory for B12 [17]. It is, however, not clear whether B12 transporters and receptors regulate hepatic B12 levels in relation to the circulating B12 levels. There is currently no data that elucidates the role of CD320 and TCN2 with intracellular levels of B12 in hepatocytes. Therefore, the present study was designed to assess and understand the relationship between circulating the B12 level, its intracellular hepatic concentrations and the levels of CD320 and TCN in both the in vitro (human) and in vivo (mouse) model of liver.

## 2. Results

### 2.1. Intracellular and Tissue Levels of B12

#### 2.1.1. Intracellular B12 Level in Hep G2 Cell Line

At low (0.025–0.1 nM) and physiological levels of extracellular B12 concentration (1 nM), the intracellular B12 levels (0.07–0.15 nmol/g protein) were relatively higher. However, supraphysiological levels of extracellular concentration of B12 (10 nM, 20 nM, 50 nM, 100 nM, 200 nM, 400 nM and 500 nM) resulted in only a slight increase in the intracellular B12 levels (i.e., 0.24, 0.45, 0.68, 0.73, 1.91, 2.33 and 3.99 nmol/g protein, respectively) (Figure 1A). Thus, we found that though there was a 20,000-fold increase in the extracellular B12 concentration (i.e., from 0.025 to 500 nM B12), the intracellular B12 level showed only 57-fold increase (i.e., from 0.07 to 3.99 nmol/g protein). Our results suggest that the efficiency of B12 transport (related to the extracellular level of B12) is reduced at supraphysiological levels of B12 (>1 nM). However, physiological levels of B12 are not accompanied by a tremendous reduction in the intracellular B12 because the transportation system has a high affinity for B12 and hence improves the rate of its uptake.

#### 2.1.2. Intracellular B12 Level in Mice Liver Tissues

To evaluate the tissue-specific relative uptake of B12, we assessed the levels of B12 in extracts of liver tissue relative to plasma. The average B12 concentration in the plasma of mice with low B12 (LB (*n* = 6)) was 0.82 nM (physiological level) and the tissue extract was 18 nmol/g protein. In contrast, the average concentration in the high plasma B12 (HB (*n* = 6)) group was 5.36 nM (supraphysiological level) and the tissue extract level of B12 was 34 nmol/g protein (Figure 1B). Like the in vitro model, we observed whilst there was a 6.5-fold increase (i.e., 0.82–5.36 nM) in the plasma B12 concentration, the tissue extract resulted in only a 2-fold increase (i.e., 18–34 nmol/g protein) in B12 concentration.

### 2.2. B12 Transcobalamin Receptor (CD320)

#### 2.2.1. Hep G2 Cells

We assessed the protein and mRNA expression of CD320 in the Hep G2 cell line at ten different B12 media concentrations. In the experiment with low and physiological levels of extracellular B12 concentrations (0.025–1 nM), the gene expression of the CD320 receptors was significantly higher compared to the control (500 nM). At supraphysiological levels of B12 media concentrations (10 nM, 20 nM, 50 nM, 100 nM, 200 nM and 400 nM), however; there was a decrease in the gene expression of transcobalamin receptor (CD320), with no significant difference from the control (extracellular B12 = 500 nM) (Figure 2A). Similarly, at media concentrations of B12 (such as 0.025 nM, 0.1 nM, 10 nM and 20 nM), we observed upregulation of CD320 protein levels compared to the control. The highest CD320 protein expression was observed in cells cultured at the lowest B12 media concentration (0.025 nM). However, cells cultured at supraphysiological levels of B12 media concentrations (such as 50 nM, 100 nM, 200 nM and 400 nM) had decreased CD320 protein levels with no significant difference from the control (500 nM) (Figure 2B).

#### 2.2.2. Mice Liver

Then, we assessed the mRNA expression and protein levels of CD320 in the liver tissue extracts of mice with high and low levels of B12 (HB and LB, respectively). There was no significant difference in the CD320 mRNA expression between the LB and HB mice (Figure 2C). However, we observed that the CD320 protein levels in the tissue extracts of LB mice (at physiological levels) were significantly increased compared to the HB group (supraphysiological levels) (Figure 2D). This suggests that although the difference in expression of B12 receptors was not obvious at the transcript level, the protein expression was significantly higher in the LB group implying that modifications could have occurred at the post-translational level in mice liver.

### 2.3. B12 Transporter Transcobalamin II (TCN2)

#### 2.3.1. Hep G2 Cells

To further evaluate the transport of B12, we assessed the protein and mRNA expression of B12 transporter transcobalamin II (TCN2) in Hep G2 cell line cultured in ten different B12 media. At media B12 conditions (0.025–0.1 nM), mRNA expression of hepatic TCN2 was significantly higher. In contrast, we observed that mRNA expression of TCN2 was decreased as concentrations of B12 increased (such as 1 nM, 10 nM, 20 nM, 50 nM, 100 nM, 200 nM and 400 nM), showing no significant difference from the control. The highest TCN2 mRNA expression was observed at the lowest B12 media concentration (0.025 nM) compared to the control (Figure 3A).

Similarly, cells cultured at media concentrations of B12 (0.025–50 nM) resulted in high TCN2 protein expression compared to the control. However, supraphysiological levels B12 media concentrations (such as 100 nM, 200 nM and 400 nM) showed low levels of TCN2 protein expression that were not significantly different from the control (Figure 3B).

#### 2.3.2. Mice Liver

Like with Hep G2 cells, we observed that both mRNA expression and protein levels of TCN2 were significantly higher in the tissue extracts of mice with a low plasma B12 (LB) compared to the HB group. The LB group (at physiological levels of B12) showed the expression of mRNA and protein of TCN2 to be 75% (Figure 3C) and 50% (Figure 3D) higher respectively, compared to the HB mice.

## 3. Discussion

In this study, both the in vitro and in vivo model demonstrate that physiological extracellular B12 concentrations led to an increased relative uptake of B12 than supraphysiological B12 levels. Both gene and protein expression of B12 receptor CD320 and transporter TCN2 also increased at physiological B12 concentrations. These results thus imply that physiological levels of B12 in the extracellular medium (or blood plasma) induce hepatocytes to accelerate the expression of B12 receptor CD320 and transporter TCN2, and cellular uptake of B12.

Both higher and lower concentrations of circulating B12 are common. For instance, B12 deficiency may be caused by vegetarianism, prolonged metformin treatment in type 2 diabetes patients and during pregnancy [18,19], whereas high plasma levels may result from ingestion of B12 supplement [20]. B12 levels range from low (0.025–0.1 nM), physiological (0.1–1 nM) and supraphysiological (>1 nM) levels. In the current study, we determined the intracellular levels of B12 in Hep G2 cells cultured in ten different B12 media to assess how liver cells may respond to lower, physiological and supraphysiological levels in humans. We found that the increasing extracellular B12 does not stimulate an indefinite or a dose-dependent increase of the intracellular B12 because of a limited capacity of the transportation system, which becomes saturated and then downregulates its rate. The saturable character of B12 uptake was also found in the intestine [21], though up- and downregulation of transporters was not demonstrated. A study in the rat model showed that several small doses of dietary B12 are more efficient in increasing blood and tissue B12, than a single large dose [11]. In a human study, Yajnik et al. showed that supplementation with a physiological dose of oral B12 over 11-months in asymptomatic low B12 adolescent women significantly improved the B12 levels in these women. [22]. A recent study in early postpartum women in urban Tanzania showed that B12 was low in breast milk and was not significantly increased by high dose supplementation [23]. Our data may also suggest that the supraphysiological levels of B12 in blood induces only a moderate to low increase in the intracellular B12 and may not be required for further cellular uptake.

Importantly, we observed increased protein and gene expression of the B12 receptor (CD320) in Hep G2, particularly at physiological circulating B12 levels. In contrast, supraphysiological B12 media concentrations resulted in a lower expression of protein CD320, though there was no significant difference in the mRNA expression, revealing that mRNA expression of a gene does not always predict its protein level [24]. Our findings are in line with an earlier study in hepatocytes, which showed that modulations in the cellular uptake of B12 was dependent on the phases of the cell cycle and alterations in the number of cellular receptors rather than affinity [25]. In this study, we also observed higher TCN2 levels in both Hep G2 cell line and in mice under decreasing concentrations of extracellular B12 and vice versa. B12 transporter—TCN2 is principally produced by the kidneys and hepatocytes, intestines, endothelia and monocytes [26,27,28]. De novo biosynthesis of TCN2 has been demonstrated in the liver perfusates and cultured liver parenchymal cells using rat models [29,30], supporting our observation. Although TCN2 accounts for approximately 22–37% of plasma B12 [31], it is the key transporter of up to 99% of active B12 (holo-TC) into the tissues due to its fast turnover [32]. Therefore, modulations in both gene and protein expression of TCN2 in the liver reflects its role in the regulation of B12 uptake and metabolism. Supporting our findings, an animal study showed that low B12 increased the expression of TCN2 in the maternal and foetal tissues [33]. Clinically, low plasma B12 may be observed during pregnancy due to increased demand of B12 for foetus [34], and some perturbation in the receptor or/and transporter might be expected. For example, a recent study showed a progressive increase in serum levels of soluble CD320 receptors up to 35 weeks of gestation [35]. High levels of serum sCD320 receptors during pregnancy might be derived from the placenta, where higher expression of the receptor has been shown [35]. Taken together, our data highlight that the intracellular B12 levels could be maintained by upregulation of B12 uptake at its limited supply and downregulation at an excessive supply. All in all, such mechanisms may shunt the excess of B12 out of its cells for mobilization to other tissues where it is most needed. In case of B12, the liver and kidneys are typical organs that dramatically change their intracellular levels depending on extracellular supply of B12 and the requirement of the cells [9]. It should be, however noted, that different forms of B12 (e.g., cyano- vs. hydroxo-B12) can give different ratios of intracellular/extracellular vitamin if administered at equal quantities [9]. Possible explanation may be that the different conversion rates of the inactive form of B12 to the active form, where the unprocessed inert B12 forms are expelled [9].

## 4. Materials and Methods

### 4.1. Cell Culture

The cell culture was performed with slight modifications, as described elsewhere [36]. Briefly, the HepG2 cell line was cultured using custom-made B12 deficient Eagles’ minimal essential medium (EMEM). The medium was supplemented with 10% foetal bovine serum (FBS), 1% penicillin/streptomycin, 1% L-glutamine and placed under 37 °C incubation with 5% CO_2_ saturation. At 90% confluence, the cells were trypsinized and seeded in 6-well plates at 75,000 cells per well and incubated with a range of different concentrations of B12 in the media, i.e., 500 nM (control), 400 nM, 200 nM, 100 nM, 50 nM, 20 nM, 10 nM, 1 nM, 0.1 nM and 0.025 nM B12. The media in each well was changed every 48 h. When the cells reached 90–100% confluence, they were harvested and stored at −80 °C for further protein and gene expression analysis.

### 4.2. Animal Model

All animal procedures were carried out in accordance with the United Kingdom Animals (Scientific Procedures) Act 1986, approved by the local ethics review committee at the University of Southampton and conducted under Home Office Project Licence number 70/6457, granted on 10 May 2017. Female C57/BL6J mice were maintained under a 12 h light/dark cycle (lights on at 07.00 h), and at a constant temperature of 22 ± 2 °C with food and water available ad libitum. The mice (*n* = 6 per group) were fed either a control diet containing 7.49 µg/kg vitamin B12 (SDS RM1 diet, LBS-Serving Biotechnology Ltd, Surrey, UK) or a high vitamin B12 diet containing 30.82 µg/kg vitamin B12 (SDS 824053 Diet, LBS-Serving Biotechnology Ltd, Surrey, UK) for 8–10 weeks. The animals were killed by cervical dislocation. The blood and liver tissue samples were collected. Plasma was extracted from the K_2_-EDTA blood samples and stored at −20 °C, and liver samples were immediately placed on ice and carefully diced, followed by washing in Hanks balanced salt solution (HBSS, Merck Life Science UK Ltd, Dorset, UK) until all blood was removed. Samples were subsequently snap frozen in liquid nitrogen and stored at −80 °C for further analysis.

### 4.3. RNA Isolation and Quantitative Real Time-PCR (qRT-PCR)

Total RNA in Hep G2 cells and mice liver tissues of different concentration of B12 were isolated using Qiazol (Qiagen, Manchester, UK) according to the manufacturer’s instruction. RNA quantification, quality and integrity (260/230 and 260/280 ratios and concentrations) were carried out using a NanoDrop spectrophotometer (Labtech, Heathfield, UK). Total RNA (300 ng) was reverse transcribed to cDNA using reverse M-MLV transcriptase (ThermoScientific, Loughborough, UK). Gene expression of CD320 in the Hep G2 cell line (*CD320*: cat no Hs00213164_m1; *TCN2*: cat no Hs00165902_m1; Applied Biosystems, Thermo Fisher Scientific, Horsham, UK) and mice liver tissues (CD320: forward primer—GGTCCAAGTCTCCGGCTCTA-, reverse primer–AGCACATGACTCAATCCTACAGT; TCN2: forward primer—CTTTGCTGGATCTTCCTTGG, reverse primer—TCCTGGGGTTTGTAGTCAGC; Merck Life Science UK Ltd, Dorset, UK) were determined by quantitative real time polymerase chain reaction (Applied Biosystems 7500 Fast Real-Time PCR Thermal Cycler, Thermo Fisher Scientific, Horsham, UK). The housekeeping genes, 18s rRNA in the HepG2 cell line (cat no 4319413E; Applied Biosystems, Thermo Fisher Scientific, Horsham, UK) and L19 in mice liver tissues (forward primer—GGAAAAAGAAGGTCTGGTTGGA, reverse primer—TGATCTGCTGACGGGAGTTG; Merck Life Science UK Ltd, Dorset, UK) were used to determine relative gene expression.

### 4.4. Western Blot Analysis

Hep G2 cells in plates were washed twice with ice-cold phosphate-buffered saline (PBS) followed by extraction buffer (RIPA—radioimmunoprecipitation assay buffer) containing phosphatase and protease inhibitors. After lysis, insoluble materials were removed by centrifugation and the supernatant was transferred to a new tube and kept at −80 °C for further analysis [18]. Similarly, proteins were extracted from mouse liver tissues by adding ice-cold RIPA buffer to the tube containing the tissue and then homogenised. Following centrifugation, the supernatant was transferred to a fresh tube and kept at −80 °C for further analysis. Protein content in the samples was quantified using the Bradford assay (Bio-Rad Laboratories Ltd, Watford, UK). Western blots were carried out from both Hep G2 cells and liver tissues using the protocol described elsewhere [19].

### 4.5. Measurement of B12

Hep G2 cells were washed twice with ice-cold phosphate-buffered saline (PBS). The cells were then harvested or lysed in PBS and stored at −80 °C. Mice liver extracts were obtained by homogenising the liver tissues in PBS and stored at −80 °C. Intracellular concentrations of B12 in Hep G2 cell lysates, liver tissues and plasma were determined by electrochemiluminescent (ECL) immunoassay kit (compatible to blood plasma, tissue and cellular levels of B12) using a Roche Cobas immunoassay analyzer (Roche Diagnostics Ltd, West Sussex, UK). In cell culture media without supplemented B12, foetal bovine serum (10% FBS) contributed less amounts of B12 (28 pM) to the media, which is well below the physiological level (0.15–1 nM). The measuring range of the B12 kit was 0.037–1.476 nM. Some of the cell line and tissue samples were in nanometre levels of B12, therefore, samples with B12 concentrations above the measuring range were diluted manually (several folds) with Diluent Universal. After dilution, the results were multiplied by the dilution factor. Following the manufacturer’s recommendation, a linearity assay was performed using a known low analyte containing plasma as standard. The final measurements were obtained in triplicates for every sample (*n* = 6 for each experimental condition) to ensure reproducibility, and B12 concentrations were obtained by multiplying the average by the dilution factor.

### 4.6. Statistical Analysis

Results are presented as mean ± standard error of the mean (SEM) for in vitro experiments (*n* = 6) and in vivo samples (*n* = 6). Data for all samples obtained were tested for normality, prior to analysis, with the Kolmogorov–Smirnov and Shapiro–Wilk normality tests using Prism 8 (GraphPad Software, San Diego, CA, USA). Differences between either parametric groups or non-parametric groups were observed respectively by performing Student’s *t*-test or Mann–Whitney U test. *p* values of <0.05 were considered statistically significant.

## 5. Conclusions

In conclusion, our study highlights that when extracellular B12 levels are at physiological levels, the intracellular B12 levels are higher, which is supported by the increased gene and protein expression of receptor/transporter in both in vitro and in vivo models. This shows that at physiological B12 concentration, the active transport of B12 in the tissues accelerate. Our evidence thus validates the long-existing hypothesis that the cells indeed, regulate the uptake of B12. However, at supraphysiological levels of B12, the intracellular concentrations are relatively reduced, the receptor/transporter get downregulated, presumably to protect the cells from long-term exposure to high levels. This supports that regular smaller doses of B12 supplementation might be better than larger, infrequent dosing of B12 in deficient individuals. In addition, overloading B12 deficient patients with supplements may not present any additional benefit but regular monitoring of patients to aim for physiological levels may be recommended. This may maintain optimal levels of the transporter/receptor system across various tissues, although this requires additional studies in other tissues.

## Figures and Tables

**Figure 1 ijms-22-03089-f001:**
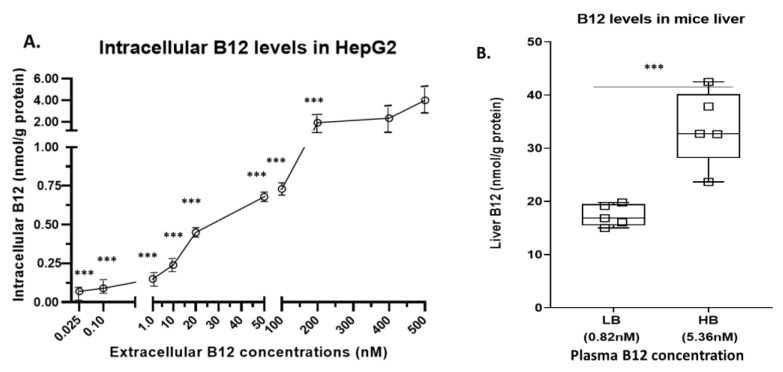
Intracellular B12 levels in Hep G2 cell line and mice liver corresponding to levels of B12 in circulation: (**A**) Hep G2 cell line was cultured in ten different concentrations of B12 (500 nM (control), 400 nM, 200 nM, 100 nM, 50 nM, 20 nM, 10 nM, 1 nM, 0.1 nM and 0.025 nM) in EMEM media. B12 levels in Hep G2 cell line lysates (*n* = 6) were measured. (**B**) Liver tissues of both high plasma B12 (HB) group (5.36 nM) and low plasma B12 (LB) group (0.82 nM) mice were homogenized in 500 µL of PBS and the B12 of lysate and plasma were measured. The data is representative of mean ± SEM (*n* = 6), * indicates significance compared to the control; *** *p* < 0.001.

**Figure 2 ijms-22-03089-f002:**
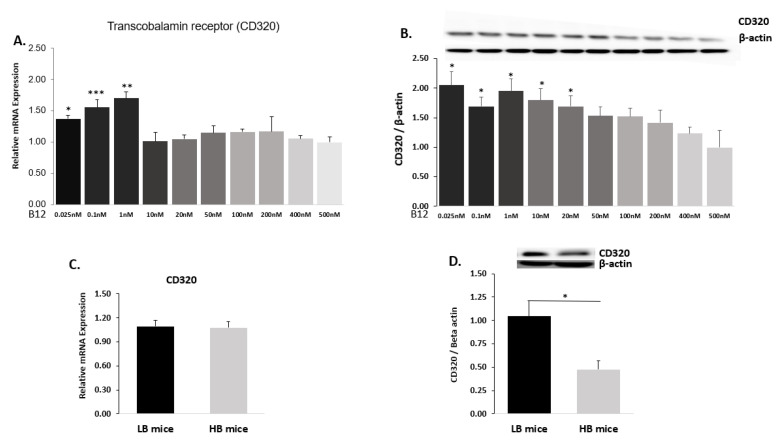
Transcobalamin receptor (CD320) mRNA expressions and protein levels in Hep G2 and liver tissues: (**A**) The mRNA expression of transcobalamin receptors (CD320) normalized to 18S rRNA endogenous control and (**B**) CD320 protein levels related to β-actin in Hep G2 cells at ten different B12 concentrations (500 nM (control), 400 nM, 200 nM, 100 nM, 50 nM, 20 nM, 10 nM, 1 nM, 0.1 nM and 0.025 nM). (**C**). The CD320 mRNA expression in liver tissues of HB and LB mice normalized to L19 endogenous control and (**D**) CD320 protein levels related to β-actin in liver tissues of HB and LB mice. The data are representative of mean ± SEM (*n* = 6) and * indicates significance compared to the control; * *p* < 0.05, ** *p* < 0.01, *** *p* < 0.001.

**Figure 3 ijms-22-03089-f003:**
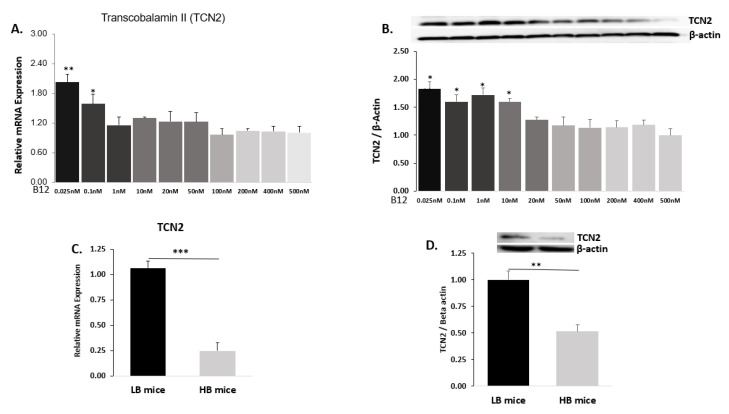
B12 transporter (transcobalamin II, TCN2) mRNA expressions and protein levels in Hep G2 and liver tissues: (**A**) The mRNA expression of TCN2 in Hep G2 normalized to 18S rRNA endogenous control and (**B**) protein level of TCN2 related to β-actin in Hep G2 at ten different B12 media concentrations (500 nM (control), 400 nM, 200 nM, 100 nM, 50 nM, 20 nM, 10 nM, 1 nM, 0.1 nM and 0.025 nM). (**C**) The TCN2 mRNA expression in liver tissues of HB and LB mice normalized to L19 endogenous control and (**D**) represents the levels of B12 transporter TCN2 protein related to β-actin in HB and LB groups of mice. The data are representative of mean ± SEM (*n* = 6), and * indicates significance compared to the control; * *p*< 0.05, ** *p*< 0.01, *** *p*< 0.001.

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
