# Peer review of "Intracellular and Tissue Levels of Vitamin B12 in Hepatocytes Are Modulated by CD320 Receptor and TCN2 Transporter"

_ijms, 2021, doi:10.3390/ijms22063089_

Round 1

Reviewer 1 Report

This manuscript is well described on a study investigating the effect of vitamin B12 loading on vitamin B12 concentration in HepG2 cells and mouse liver from the aspect of the relationship between mRNA and protein expression of CD320 and TCN2. However, I have some comments to the author.

Comment 1: In Figure 1B, the origin and the plot with LB mice are connected by a straight line. However, since the author has not measured plasma B12 concentration = 0 nM in mice, it is strange to set liver B12 concentration = 0 (actually, even if plasma B12 concentration = 0 nM, liver B12 concentration will be not 0, I think).

Comment 2: Similarly, in Figure 1B, LB mice and HB mice are connected by a straight line. However, since it is not an experimental system in which plasma B12 concentration is continuously increased, it is not correct to connect these two points with a straight line as measured continuously.

Comment 3: Regarding Figure 2C and 2D, although there is no difference in the expression levels of LB mice and HB mice CD320 mRNA, but there is a difference in protein expression. Appropriate explanation should be given for this difference in results.

Comment 4: It is understandable that the intracellular or liver B12 concentration fluctuates depending on the Extracellular or plasma B12 level, and the expression levels of CD320 and TCN2 change. However, it cannot be said that the author has described in the conclusion that "The efficiency of B12 uptake is either increased or decreased, when the reducing level of B12 is lower or higher respectively, thereby maintaining intracellular B12 at a relatively stable level", is it? Because, the intracellular B12 concentration shown in Figure 1 is significantly higher than the physiological level of intracellular B12 levels (0.07 – 0.15 nmol / g protein) stated at the beginning of the Result, whereas the CD320 The changes in the expression level of TCN2 and TCN2 are not so remarkable, and it seems that there is no effect and fluctuation that can be said to regulate the intracellular B12 level.

Author Response

RESPONSE TO REVIEWERS

Reviewer 1:

Comment 1. In Figure 1B, the origin and the plot with LB mice are connected by a straight line. However, since the author has not measured plasma B12 concentration = 0 nM in mice, it is strange to set liver B12 concentration = 0 (actually, even if plasma B12 concentration = 0 nM, liver B12 concentration will be not 0, I think).

Response: We thank the reviewer for pointing this out. Figure 1B has been corrected accordingly.

Changes made in the manuscript and location: Figure 1B replaced with corrected version

Comment 2. Similarly, in Figure 1B, LB mice and HB mice are connected by a straight line. However, since it is not an experimental system in which plasma B12 concentration is continuously increased, it is not correct to connect these two points with a straight line as measured continuously

Response: As reviewer suggested, we have corrected Figure 1B accordingly.

Changes made in the manuscript and location: Figure 1B is replaced with corrected version

Comment 3: Regarding Figure 2C and 2D, although there is no difference in the expression levels of LB mice and HB mice CD320 mRNA, but there is a difference in protein expression. Appropriate explanation should be given for this difference in results.

Response: As reviewer suggested, we have given further explanation in the manuscript.

Changes made in the manuscript and location: Page 6, line 221-223; page 8, line 293-296.

Comment 4: It is understandable that the intracellular or liver B12 concentration fluctuates depending on the Extracellular or plasma B12 level, and the expression levels of CD320 and TCN2 change. However, it cannot be said that the author has described in the conclusion that "The efficiency of B12 uptake is either increased or decreased, when the reducing level of B12 is lower or higher respectively, thereby maintaining intracellular B12 at a relatively stable level", is it? Because, the intracellular B12 concentration shown in Figure 1 is significantly higher than the physiological level of intracellular B12 levels (0.07 – 0.15 nmol / g protein) stated at the beginning of the Result, whereas the CD320 The changes in the expression level of TCN2 and TCN2 are not so remarkable, and it seems that there is no effect and fluctuation that can be said to regulate the intracellular B12 level.

Response: We acknowledge and appreciate the recommendation. We have addressed and modified our conclusion accordingly.

Changes made in the manuscript and location: Page 8-9, line 332-340.

Reviewer 2 Report

Manuscript ID: ijms-1137377

Title: Intracellular and tissue levels of vitamin B12 in hepatocytes are modulated by CD320 receptor and TCN2 transporter

Authors: Joseph Boachie, Adaikala Antonysunil*, Ilona Goljan, Jinous Samavat, Felino R. Cagampang, Ponnusamy Saravanan*

This manuscript tried to demonstrate the interrelationship between intrahepatic and circulating B12 levels utilizing in vitro (HepG2 cells) and in vivo (mouse liver tissues/plasma) models. The authors attempted to suggest that the gene expression and/or protein levels of CD320 and TCN2 could involve in the efficacy of B12 transport into hepatocytes. It could impact B12 deficiency-related human disease models to provide information that the efficient dosage of B12 concentration. However, overall data presentation, statistical analysis, and the interpretation of the results are insufficient to support the authors’ conclusion.

Strength of the manuscript:

  1. This study has the novelty to suggest the efficient dosage of B12 supplements. The authors also tried to prove the involvement of CD320 and TCN2 gene expression/protein levels in hepatocytes.

Major concerns:

  1. The rationale for using HepG2 cells is obscure. The HepG2 cells are liver cancer cell lines, which could have different gene/posttranslational regulatory systems for CD320 and TCN2 than normal hepatocytes.
  2. The statistical analysis is missing in Figure 1. Besides, the interpretation of results has a discrepancy with the statistical analysis in Figures 2 and 3. In the same context, controls are ambiguous in WB and gene expression data (Figures 2 and 3).
  3. The authors should clarify how they prepared and collected mouse liver tissue extract samples for measuring the intracellular B12 level in Figure 1B. As the liver has many blood veins, the homogenization of the whole liver as intracellular tissues is not reliable. The authors should perfuse the liver to remove blood/plasma for making an exact comparison of B12 concentrations between plasma and tissue extracts.
  4. TCN2 and b-actin or CD320 and b-actin should be shown in the same blot for Western blot data in Figures 2 and 3 to clarify the different protein levels of TCN2 and CD320 at different B12 extracellular concentration.

Minor issues:

Line 186-199: The format is different from the other parts of the manuscript. Corrections needed.

Reviewer 3 Report

The manuscript determines the role of CD320 receptor and TCN2 transporter in vitamin B12 modulation in hepatocytes. The authors found that levels of B12 in the extracellular media or circulation accelerate active transport of B12, and expression of CD320 and TCN2, resulting in higher relative uptake of B12 in hepatocytes. The author presents a nice piece of work that will help in understanding the metabolism of vitamin B12. While this study provided some interesting data, there are few issues to be carefully addressed.

  • The title and abstract reflect the content of the work.
  • The introduction section requires some improvement. Please add the rationale for using the CD320 receptor and TCN2 transporter.
  • Please do some correlation and regression statistical analysis to find an association between CD320 receptor, TCN2 transporter, and vitamin B12.

Round 2

Reviewer 1 Report

Thank you for answering to my comments and correcting the Figure 1B.

Reviewer 2 Report

None